# Enhancing Sustainability in Potato Crop Production: Mitigating Greenhouse Gas Emissions and Nitrate Accumulation in Potato Tubers through Optimized Nitrogen Fertilization

Camila Seno Nascimento, Carolina Seno Nascimento, Breno de Jesus Pereira , Paulo Henrique Soares Silva, Mara Cristina Pessôa da Cruz and Arthur Bernardes Cecílio Filho *

Department of Agricultural Production Sciences, São Paulo State University, Jaboticabal 14884-900, Brazil; camilaseno@gmail.com (C.S.N.); senocarolina@gmail.com (C.S.N.); brenojp93@gmail.com (B.d.J.P.); phsoares18@yahoo.com.br (P.H.S.S.); mcp.cruz@unesp.br (M.C.P.d.C.)
* Correspondence: arthur.cecilio@unesp.br

**Abstract:** The complex ramifications of global climate change, which is caused by heightened concentrations of greenhouse gases in the Earth's atmosphere, are deeply concerning. Addressing this crisis necessitates the immediate implementation of adaptive mitigation strategies, especially within the agricultural sector. In this context, this study aimed to assess how the supply of nitrogen (N) (0, 70, 140, and 210 kg N ha$^{-1}$) in the forms of ammonium nitrate and urea affects the agronomic performance, food quality, greenhouse gas emissions (GHG), and carbon footprint of potato plants. The examined hypothesis was that by precisely calibrating N doses alongside appropriate sourcing, over-fertilization in potato cultivation can be mitigated. A decline in stomatal conductance and net photosynthetic rate disturbs physiological mechanisms, reflecting in biomass production. Application of 136 kg N ha$^{-1}$ as urea showed a remarkable yield increase compared to other doses and sources. The highest nitrate content in potato tubers was achieved at 210 kg N ha$^{-1}$ for both sources, not exceeding the limit (200 mg kg$^{-1}$ of fresh mass) recommended for human consumption. The lowest carbon footprint was obtained when 70 kg N ha$^{-1}$ was applied, around 41% and 26% lower than when 210 kg N ha$^{-1}$ and 140 kg N ha$^{-1}$ were applied, respectively. The results demonstrated that over-fertilization not only worsened the yield and tuber quality of potato plants, but also increased greenhouse gas emissions. This information is valuable for establishing an effective fertilization program for the potato crop and reducing carbon footprint.

**Keywords:** ammonium nitrate; CO$_2$ equivalent; global warming potential; *Solanum tuberosum*; urea



## 1. Introduction

The global agriculture sector faces an unprecedented challenge in the form of climate change, marked by rising temperatures, unpredictable precipitation patterns, and the increasing concentration of greenhouse gases (GHGs) in the atmosphere. As agriculture both contributes to and is vulnerable to climate change, there is a pressing need to explore sustainable practices that not only ensure food security but also mitigate environmental impacts [1–3].

Potatoes (*Solanum tuberosum*), a staple worldwide, play a crucial role in ensuring global food security. With their versatility and nutritional value, potatoes significantly contribute to food security and serve as a vital source of sustenance for millions of people worldwide. However, conventional agricultural practices associated with potato cultivation, particularly N fertilization, contribute significantly to GHG emissions, mainly in the form of nitrous oxide (N$_2$O), a potent and long-lasting GHG that is approximately 298 times more effective at trapping heat than carbon dioxide [4]. As the demand for sustainable food production systems rises, research on potatoes has extended beyond profitability goals to focus on optimizing N fertilization approaches [5–7]. This involves matching N

supply with crop demand to reduce environmental degradation, including nitrate leaching, nitrous oxide emissions, and ammonia volatilization, as well as addressing food safety issues associated with nitrate accumulation in potato tubers [8–10].

Producing safe potato tubers under an N over-fertilization strategy is a serious problem, since potato plants tend to store large amounts of nitrate in this situation, which can be detrimental to human health [11,12]. Under excessive N supply, plants uptake nitrate above their assimilation capacity, leading to the accumulation of the element in plant tissues [13]. Nitrate, when consumed by humans, is converted to nitrite, promoting the oxidation of iron ions present in hemoglobin, transforming them from $Fe^{2+}$ to $Fe^{3+}$. This prevents the transportation of oxygen to the cells, causing methemoglobinemia, which can be life-threatening, especially for infants. Due to this, different guidelines regarding the maximum daily nitrate intake allowed for consumption have been proposed worldwide, such as in Germany, which accepts only tubers with less than 200 mg $NO_3^-$ $kg^{-1}$ fresh weight [14].

Establishing optimal N management is one of the most important and challenging decisions faced by farmers, as the N status in the soil is highly dynamic, making it nearly impossible to predict its availability over time. While several studies have focused on determining proper N rates for potato crops, the relationship between rates and sources is often overlooked, resulting in decreased N management efficiency [15,16]. Mineral N fertilizers are available on the market in various forms, each with significant price differences and unique characteristics. The choice of the appropriate chemical N form can influence how quickly the nutrient becomes available for the crop to utilize. Therefore, by selecting the proper N source, farmers can maximize N use efficiency.

In light of the above, this study aimed to explore alternative N management techniques that can mitigate GHG emissions without compromising crop yield and quality. The hypothesis tested herein is that the establishment of the optimum N dose, combined with the appropriate N source, can alleviate the over-fertilization practices typically employed by potato growers. This approach could lead to maximum returns on N supply while also improving potato agronomic performance, minimizing nitrate ($NO_3^-$) accumulation in tubers, and reducing greenhouse gas emissions and the overall carbon footprint.

## 2. Materials and Methods

### 2.1. Experimental Conditions

The experiment was conducted in the field at São Paulo State University (UNESP), Jaboticabal, São Paulo, Brazil (21°15′22″ S, 48°15′58″ W, and 575 m a.s.l.). The rainfall, mean maximum, mean minimum, and average temperatures during the experimental period were 48.7 mm, 28.7 °C, 18.1 °C, and 20.5 °C, respectively. The meteorological data were collected at the university's Agroclimatological Station.

The soil was classified as a typical Eutrophic Oxisol with a clayey texture [17]. The chemical properties of the pre-planting soil from the layer spanning 0–20cm in depth were pH ($CaCl_2$): 5.8, organic matter: 23 g $dm^{-3}$, P(resin): 60 mg $dm^{-3}$, K: 5.6 $mmol_c$ $dm^{-3}$, Ca: 25 $mmol_c$ $dm^{-3}$, Mg: 12 $mmol_c$ $dm^{-3}$, H + Al: 16 $mmol_c$ $dm^{-3}$, cation exchange capacity: 58.2 $mmol_c$ $dm^{-3}$, and base saturation of the soil: 73%.

### 2.2. Experimental Design and Treatments

The experiment was designed in a complete randomized block, in a 2 × 4 factorial scheme, with four replicates. The treatments consisted of two N sources (ammonium nitrate and urea) and four N rates at side-dressing (0, 70, 140, and 210 kg N $ha^{-1}$) applied 3 cm apart along the rows at the beginning of tuber formation, before hilling.

Each experimental plot consisted of a 5 m long row with 0.80 m spacing between rows, and each containing fourteen potato plants. Data were collected from the ten central plants, disregarding the two plants at each end.

*2.3. Experimental Setup*

The experimental site was prepared via plowing and harrowing before opening furrows for planting, fertilization, and distribution of seed tubers. According to chemical analysis, it was not necessary to perform liming before planting, since the soil base saturation was above 60% [18]. At planting, fertilization consisted of 1 kg B ha$^{-1}$ (boric acid), 100 kg $K_2O$ ha$^{-1}$ (potassium chloride), 200 kg $P_2O_5$ ha$^{-1}$ (simple superphosphate), and 40 kg N ha$^{-1}$ (using the source established in the treatments), applied in the planting furrows at an average depth of 0.15 m. Subsequently, whole seed tubers of the cultivar Agata were distributed 0.05 m above and spaced 0.35 m apart in the furrows. The seed tubers used in the experiment had an average weight and diameter of 76 g (ranging from 69 to 94 g) and 38 mm (ranging from 33 to 46 mm), respectively.

In the side-dressing fertilization, N was supplied at the rates and sources established in the treatments, 30 days after planting (DAP).

Weed control was carried out via manual hoeing, and pests and diseases were prevented using phytosanitary products registered for the potato crop. Irrigation was performed by sprinkler at a rate of 5 mm day$^{-1}$.

The harvest was carried out at 80 DAP.

*2.4. Characteristics Evaluated*

To assess the nutritional status of the potato plants, the third leaf from the apical tuft (one per plant of 10 total plants) was collected at 40 DAP, as described by Lorenzi, Monteiro, and Miranda Filho [19]. Subsequently, the collected samples were washed, dried, ground, digested, and analyzed to determine the N, P, and K contents (g kg$^{-1}$), following the procedure outlined by Miyazawa et al. [20].

Gas exchange parameters were determined for the upper third of the newly developed leaves of potato plants. The internal $CO_2$ concentration (Ci; µmol $CO_2$ m$^{-2}$ s$^{-1}$), transpiration rate (E; mmol $H_2O$ m$^{-2}$ s$^{-1}$), stomatal conductance (gs; mmol $H_2O$ m$^{-2}$ s$^{-1}$), and net photosynthetic rate (A; µmol m$^{-2}$ s$^{-1}$ $CO_2$) were evaluated using a portable gas exchange device (LC-PRO+, ADC Bioscientific Ltd. Hoddesdon, UK) under a photosynthetically active radiation (PAR) of 1200 µmol m$^{-2}$ s$^{-1}$. Based on these data, water use efficiency (WUE; µmol $CO_2$ mmol$^{-1}$ $H_2O$) was calculated through A/E [21].

Evaluations of total chlorophyll (Chl *a* + *b*) and carotenoids were determined spectrophotometrically at 40 DAP. Fresh tissue samples collected from the upper third of the first newly developed leaf of potato plants were added to tubes with 1.5 mL acetone (80%). After shaking for 48 h at 4 °C, the samples were read at 470, 645, and 662 nm for carotenoids, chlorophyll *a*, and chlorophyll *b*, respectively. The pigment contents were estimated according to Lichtentlaler [22], with results expressed in µg g$^{-1}$.

The dry biomass accumulation (g per plant) in the entire plant was calculated by summing the individual amounts of shoot and tuber dry biomass. The percentage distribution of dry biomass between potato constituent parts was determined by relating the dry biomass of each organ (shoot and tubers) to the total dry biomass (whole plant) [23].

The N accumulation in the shoots and tubers was determined at 73 DAP (two plants per plot). Subsequently, these data were used to determine the N accumulation (g per plant) in the whole plant (shoots and tubers) using the formula: N accumulation = ((N content g kg$^{-1}$) × (Potato dry biomass g per plant))/1000. N use efficiency was calculated using the formula: (total dry biomass accumulation)$^2$/(total N content in the plant), as described by Siddiqi and Glass [24].

The nitrate content in the tubers was appraised following the method proposed by Mantovani et al. [25]. The results obtained for dry biomass were converted into mg $NO_3^-$ kg$^{-1}$ in fresh mass.

To evaluate the impact of the boiling process on nitrate content, 100 g of homogeneous potato tuber pieces were immersed in 800 mL of boiling water for 15 min. Then, shortly after, to halt the cooking process, the samples were rapidly cooled in 800 mL of water and

ice for 7 min. Afterward, the samples were drained off (65 °C ± 5) until reaching constant masses, and they were then prepared for nitrate determination [25].

The total yield was expressed in tons per hectare (t ha$^{-1}$).

Greenhouse gas emissions (GHG) and carbon footprint were calculated with a dual focus on one functional unit: 1 kg of tubers produced. The established limits included direct and indirect GHG emissions linked to the cultivation phase of potatoes and the manufacture of inputs within the system, representing the entirety of the lifecycle from cradle to farm gate. The inputs evaluated were N fertilizer (with variations based on treatments), P and K fertilizers, limestone, pesticides (herbicide, fungicide, insecticide), irrigation electricity, and fuel (diesel—used for plowing, harrowing, bedding, limestone, and fertilizer application).

The GHG emissions calculation employed the methodology outlined by the Intergovernmental Panel on Climate Change [26], along with additional factors (Tier 2). The carbon equivalent ($CO_2$eq) of the total GHG emitted was calculated. The global warming potential of carbon dioxide ($CO_2$), methane ($CH_4$), and nitrous oxide ($N_2O$) over a 100-year period was considered as 1, 27.9, and 273, respectively [26].

The carbon footprint required to produce 1 kg of tubers was determined by dividing the GHG emissions in each treatment and repetition (kg $CO_2$ eq ha$^{-1}$ cycle$^{-1}$) by the corresponding yield of the treatment (kg ha$^{-1}$ cycle$^{-1}$).

The quantity of each input was the same for all treatments; only the amount of N fertilizer applied varied according to the treatments. A 2-year period was accounted for as the timeframe required for a new limestone application. For these inputs and materials, values were computed on an annual basis and then divided by two (representing the total cultivation cycles within a year). This process aimed to derive the value corresponding to one production cycle (Table 1).

**Table 1.** Amount of inputs and materials (kg ha cycle$^{-1}$) used for potato production.

| Source | Unity | Total [a] |
|---|---|---|
| N fertilizer | kg ha$^{-1}$ cycle$^{-1}$ | 40–250 [b] |
| P fertilizer ($P_2O_2$) | kg ha$^{-1}$ cycle$^{-1}$ | 200.0 |
| K fertilizer ($K_2O$) | kg ha$^{-1}$ cycle$^{-1}$ | 100.0 |
| Limestone | kg ha$^{-1}$ cycle$^{-1}$ | 612.5 |
| Herbicide (i.a [c]) | kg ha$^{-1}$ cycle$^{-1}$ | 1.39 |
| Fungicides (i.a [c]) | kg ha$^{-1}$ cycle$^{-1}$ | 21.19 |
| Insecticides (i.a [c]) | L ha$^{-1}$ cycle$^{-1}$ | 3.4 |
| Diesel | L ha$^{-1}$ cycle$^{-1}$ | 286.6 |
| Electricity-Irrigation | kwh ha$^{-1}$ cycle$^{-1}$ | 1205.4 |

[a] The total corresponds to the values diluted for one cycle of cultivation. [b] Total N fertilizer varied according to the treatments. [c] Active ingredient.

The calculation of diesel consumption considered machinery utilization. In this study, $CO_2$ absorbed by plants was disregarded.

## 2.5. Statistical Analysis

The statistical analysis of the dataset was performed using the Agroestat software [27]. Multiple comparisons between treatment means were conducted using the Tukey test at a 5% level of significance. Polynomial regression analysis was performed for N rates, and the equation with the highest level of significance and coefficient of determination was selected.

## 3. Results

Nitrogen (N) and phosphorus (P) increased with the individual effect of N application rates, reaching maximum contents of 31.3 and 6.9 g kg$^{-1}$ at 135 and 117 kg N ha$^{-1}$, respectively (Figure 1a,b).

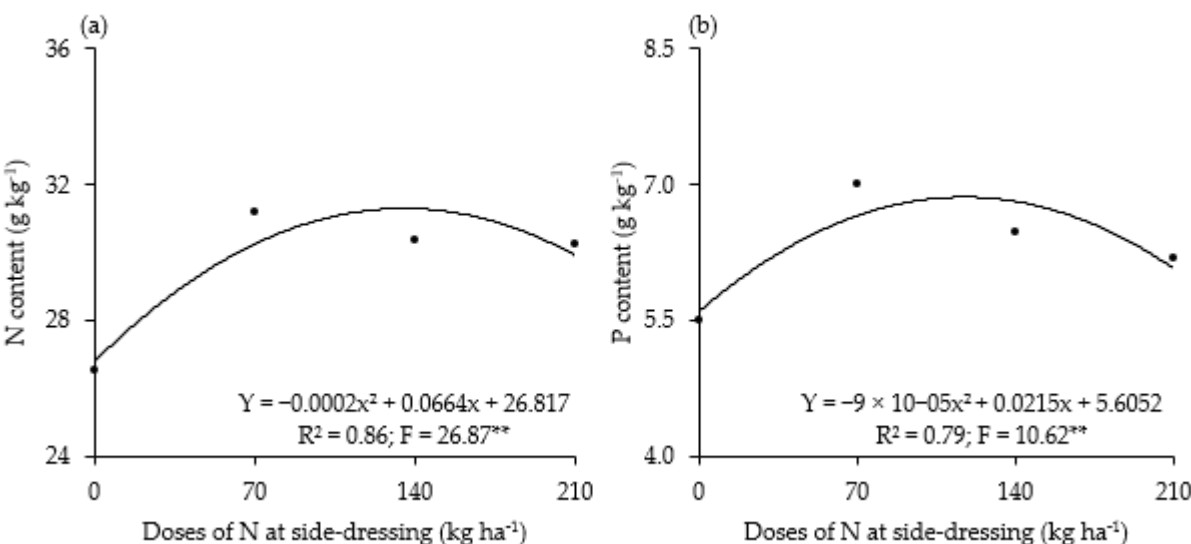

**Figure 1.** Effects of nitrogen fertilization at side-dressing on nitrogen (**a**) and phosphorus (**b**) contents in potato leaves. ** significant at the 5% and 1% probability levels by according to the F test, respectively.

Potassium (K) was unaffected by the traits evaluated, either alone or in interaction, exhibiting an average content of 31 g kg$^{-1}$.

The gas exchange parameters were significantly influenced by N management. There was a quadratic increase in stomatal conductance (gs) and net photosynthetic rate (A) as N rates increased (Figure 2c,d). Based on the regression coefficients, the estimated maximum gs (0.4 mmol H$_2$O m$^{-2}$ s$^{-1}$) and A (19.8 µmol m$^{-2}$ s$^{-1}$ CO$_2$) were achieved with the supply of 110 and 91 kg N ha$^{-1}$, respectively.

The interaction between treatments (S × D) was significant for intracellular CO$_2$ concentration (Ci), transpiration rate (E), and water use efficiency (WUE). The application of N rates as ammonium nitrate induced both Ci and E to increase following a quadratic function, reaching maximum rates of 257.4 µmol CO$_2$ m$^{-2}$ s$^{-1}$ and 3.1 mmol H$_2$O m$^{-2}$ s$^{-1}$ at 82 and 108 kg N ha$^{-1}$, respectively (Figure 2a,b). Conversely, when N was applied as urea, there was a linear increase in Ci, while no difference was observed in E rates among the treatments (Figure 2a,b). Ci in potato plants treated with urea ranged from 224.8 to 248.4 µmol CO$_2$ m$^{-2}$ s$^{-1}$.

Regarding water use efficiency (WUE), the increase in N rates using ammonium nitrate resulted in a quadratic effect on this attribute, reaching its minimum efficiency at 112 kg N ha$^{-1}$ (6.5 µmol CO$_2$ mmol$^{-1}$ H$_2$O), representing a decrease of 23.4 and 17.6% compared to untreated plants and plants at the highest N dose (210 kg N ha$^{-1}$), respectively (Figure 2e). WUE in potato plants did not show a significant difference when using urea as the N source.

The photosynthetic pigments of potato plants depend on the N rates applied throughout cultivation. A remarkable increase in both total chlorophyll (*a* + *b*) and carotenoid contents was observed in response to N application (Figure 3a,b). At the highest dose applied (210 kg N ha$^{-1}$), the total chlorophyll (*a* + *b*) and carotenoid contents were 82.6 (0.283 µg g$^{-1}$) and 43.8% (0.138 µg g$^{-1}$) higher than in untreated plants, respectively.

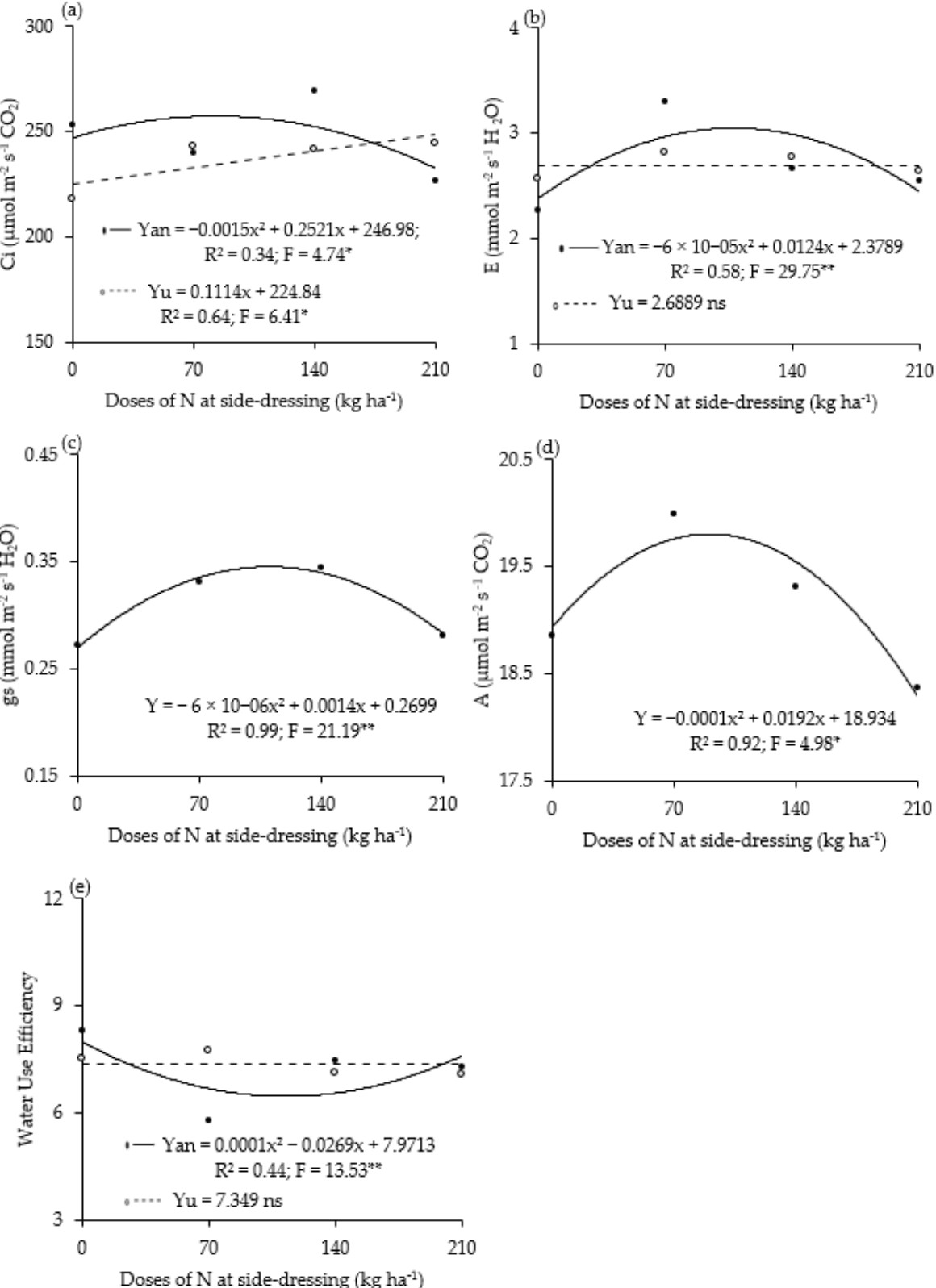

**Figure 2.** Effects of nitrogen fertilization at side-dressing on internal $CO_2$ concentration (**a**), transpiration rate (**b**), stomatal conductance (**c**), net photosynthetic rate (**d**), and water use efficiency (**e**) in potato leaves. *, ** significant at the 5% and 1% probability levels per the F test, respectively. [ns] not significant per the F test at the 5% probability.

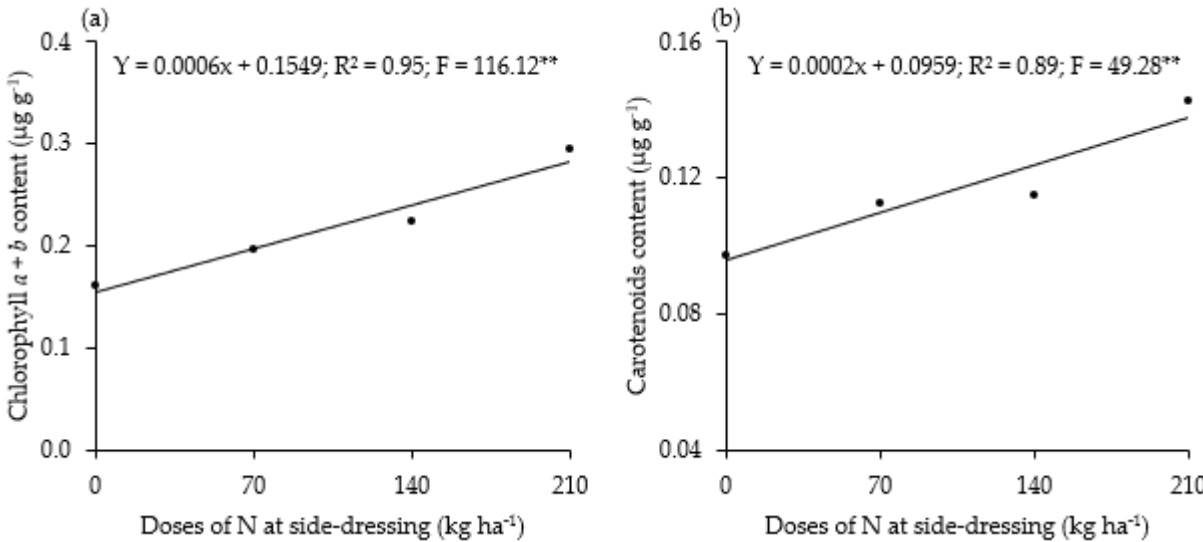

**Figure 3.** Effects of nitrogen application at side-dressing on chlorophyll *a + b* (**a**) and carotenoid (**b**) contents in potato leaves. ** significant at 5% and 1% probability levels per the F test, respectively.

The shoot dry biomass (SDM) and total dry biomass accumulation (TODA) were influenced by the interaction of N sources and rates. SDM and TODA increased quadratically with increasing N rates from ammonium nitrate (Figure 4a,c), peaking at 114 kg N ha$^{-1}$ (54.4 g) and 108 kg N ha$^{-1}$ (252.7 g), respectively. This represented an increase of 90.3 and 47.9% compared to untreated plants. Additionally, for the urea source, SDM exhibited a quadratic effect, reaching a maximum of 58.1 g at 147 kg N ha$^{-1}$, while TODA reached its highest accumulation (251.4 g) at the highest dose studied (210 kg N ha$^{-1}$) (Figure 4a,c). N fertilization led to a quadratic effect on tuber dry biomass (TDM) (Figure 4b) in potato plants.

The percentage of dry biomass in the shoots and tubers was influenced solely by N rates. N application led to a quadratic effect, increasing biomass partitioning to the shoots (BPS) up to 132 kg N ha$^{-1}$ and, similarly, a quadratic effect on the biomass partitioning to the tubers (BPT) in potato plants with increasing N rates (Figure 4d,e). At the highest dose studied, BPT contributed to nearly 78.80% of the total plant biomass, while shoots accounted for 21.20%.

N accumulation in the shoots (NAS), tubers (NAT), and the whole potato plant (TNA) showed a quadratic effect, increasing in response to N supply (Figure 5a–c), with a higher amount accumulated in the tubers than in the shoot of the plant. N fertilization significantly raised TNA up to 135 kg N ha$^{-1}$, but TNA declined as N application was further increased beyond this value.

N use efficiency (NUE) was significantly influenced by N supply. Increasing rates of N fertilizer promoted a quadratic raise in NUE (Figure 5d), ranging from 18,230.4 to 19,793.1 kg of total dry mass per kg$^{-1}$ of N absorbed.

The yield of potato plants was affected by the interaction of N sources and rates.

The increase in N rates from the ammonium nitrate source did not affect the yield of potato plants (Figure 6). On the other hand, increasing N from urea resulted in a quadratic effect on yield, reaching a maximum of 51.45 t ha$^{-1}$ at 136 kg N ha$^{-1}$. This represented an increase of 79.3 and 14.9% in comparison to untreated plants and plants fertilized with the highest dose of N, respectively.

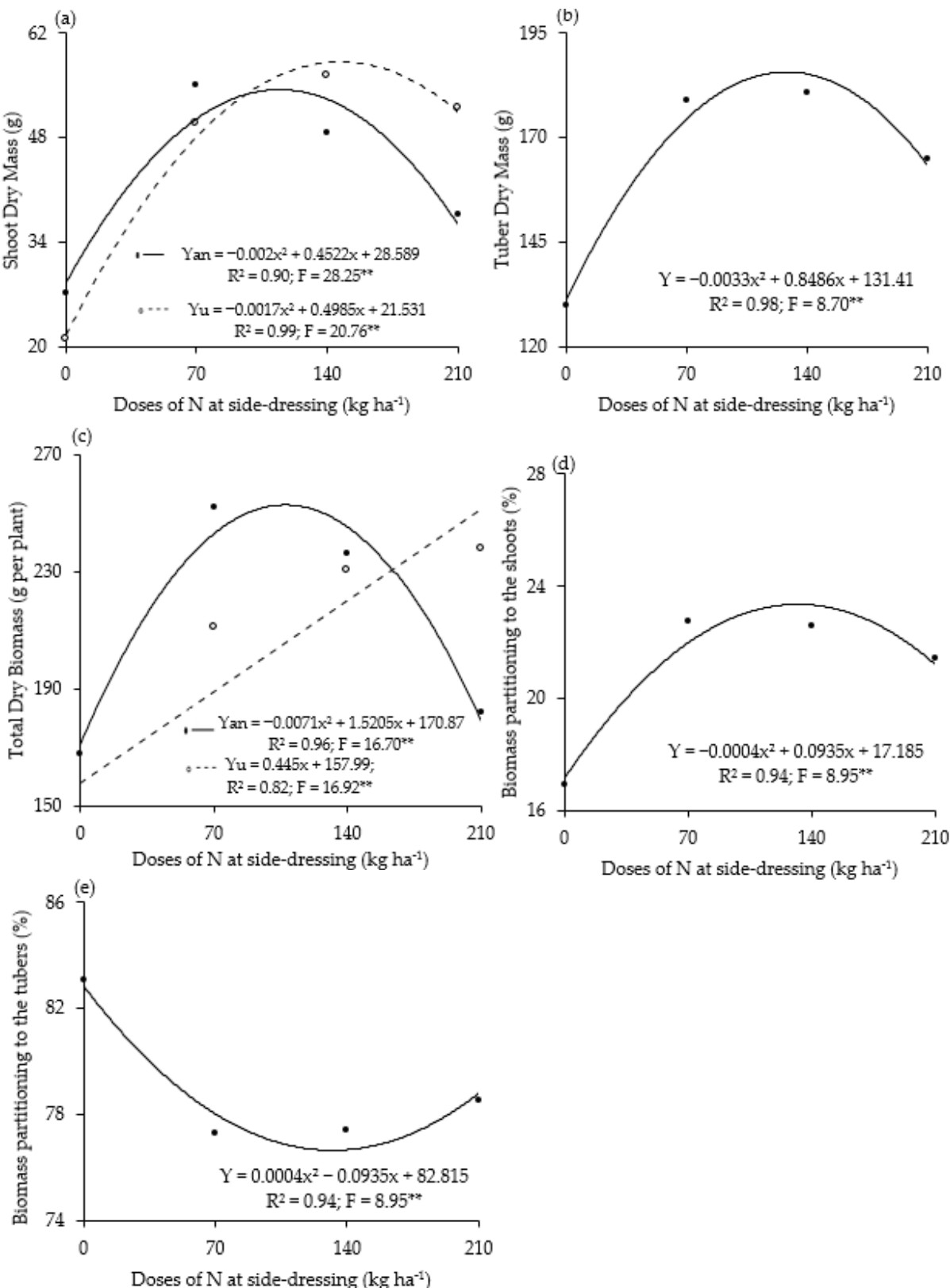

**Figure 4.** Effects of nitrogen fertilization at side-dressing on shoot dry biomass (**a**), tuber dry biomass (**b**), total dry biomass accumulation (**c**), biomass partitioning to the shoots (**d**), and biomass partitioning to the tubers (**e**) of potato plants. ** significant at 5% and 1% probability levels per the F test, respectively.

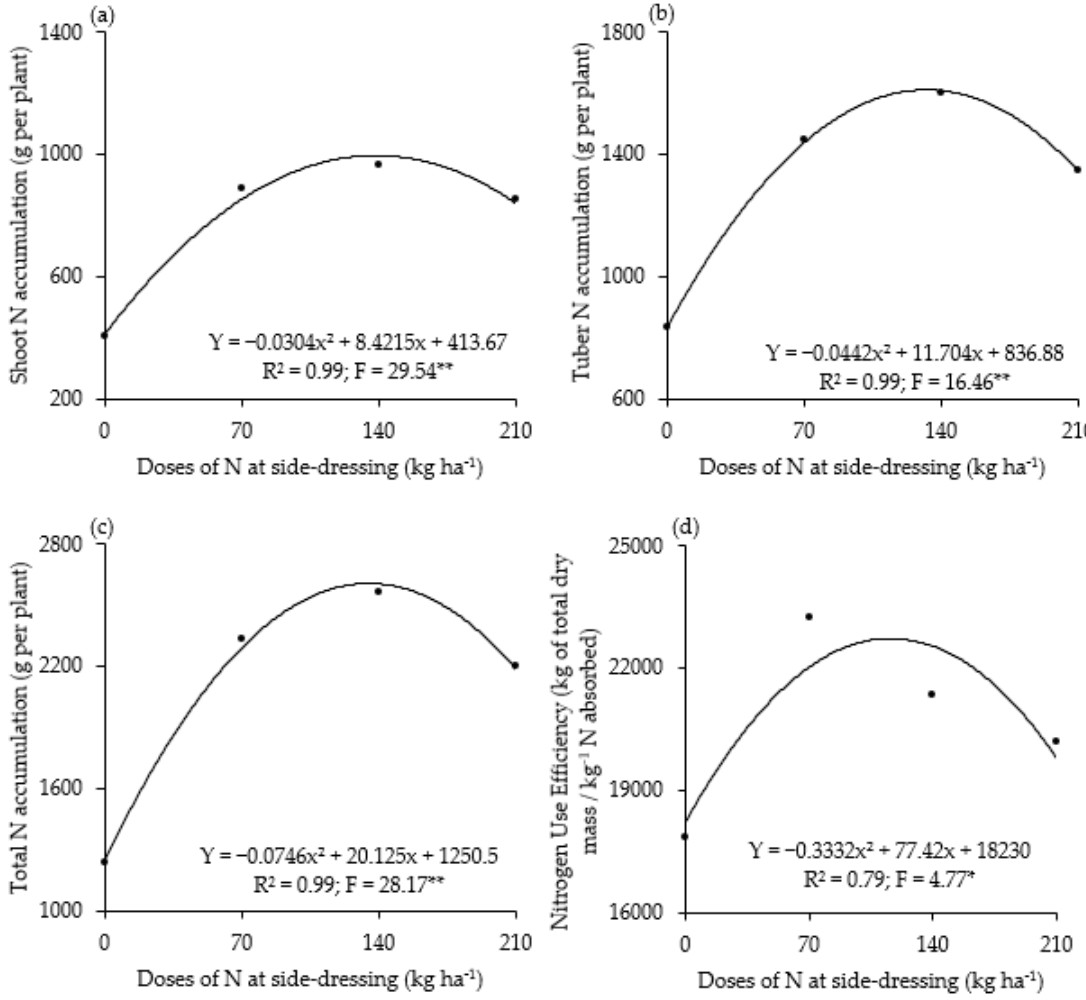

**Figure 5.** Effects of nitrogen fertilization at side-dressing on shoot N accumulation (**a**), tuber N accumulation (**b**), total N accumulation (**c**), and nitrogen use efficiency (**d**) of potato plants. *, ** significant at 5% and 1% probability levels per the F test, respectively.

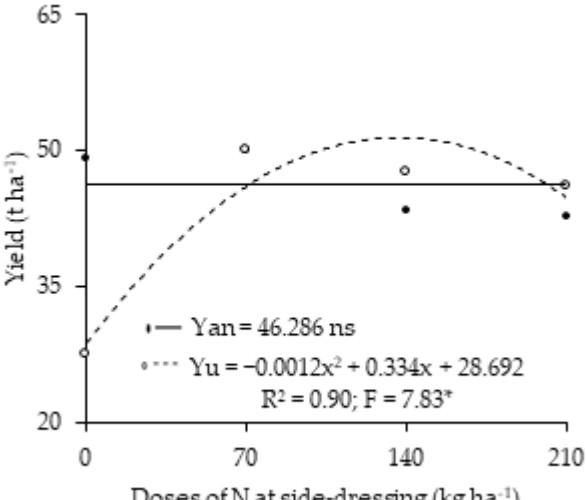

**Figure 6.** Effects of nitrogen fertilization at side-dressing on the yield of potato plants. * significant at 5% and 1% probability levels per the F test, respectively. ns not significant per the F test at the 5% probability.

The applied N sources and rates caused the nitrate content to rise at a rate following a quadratic equation, with maximums of 65.2 and 64.9 mg $NO_3^-$ $kg^{-1}$, acquired at 210 kg N $ha^{-1}$ for both sources (Figure 7a). The lowest nitrate accumulation in potato tubers was reached at 64 (44.1 mg $kg^{-1}$) and 45 kg N $ha^{-1}$ (50.8 mg $kg^{-1}$) for urea and ammonium nitrate, respectively.

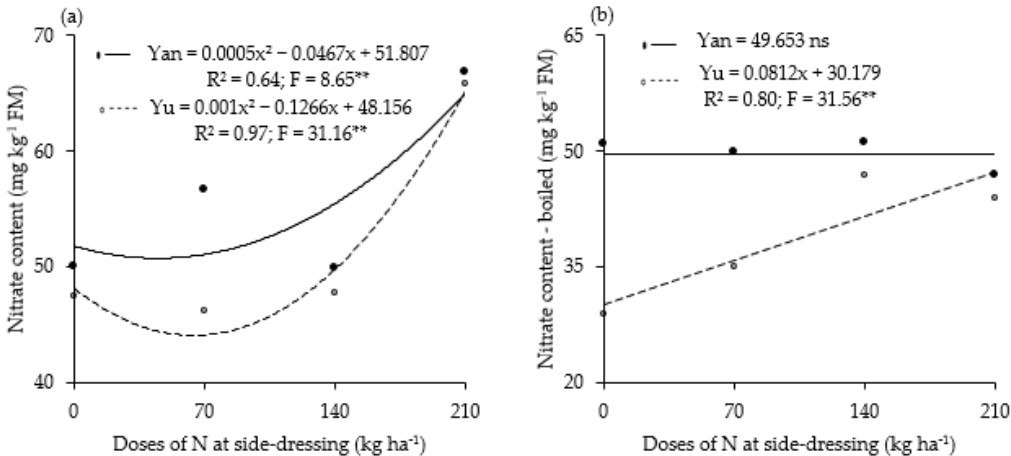

**Figure 7.** Effects of nitrogen fertilization at side-dressing on the nitrate content (**a**) and nitrate content after boiling (**b**). ** significant at 5% and 1% probability levels per the F test, respectively. [ns] not significant per the F test at the 5% probability.

A similar trend was observed for nitrate content post-boiling, which was also influenced by the interaction of N sources and rates (Figure 7b). Potato plants exhibited linear increments in this characteristic with the increase of N supply from urea, representing a rise of 56.53% (30.2–47.4 mg $kg^{-1}$) in relation to untreated plants. On the other hand, no difference was observed between the N rates sourced from ammonium nitrate.

The carbon footprint for potato production was highest when 210 kg N $ha^{-1}$ was applied, corresponding to 0.123 kg $CO_2e$ $kg^{-1}$ of potatoes, while the lowest value (0.0718 kg $CO_2e$ $kg^{-1}$ of potatoes) was obtained when 70 kg N $ha^{-1}$ was applied (Figure 8), a difference of approximately 42% less. When 140 kg N $ha^{-1}$ was applied, the carbon footprint was 0.098 kg $CO_2e$ $kg^{-1}$ of potatoes, which was not significantly different from the highest and lowest values (Figure 8).

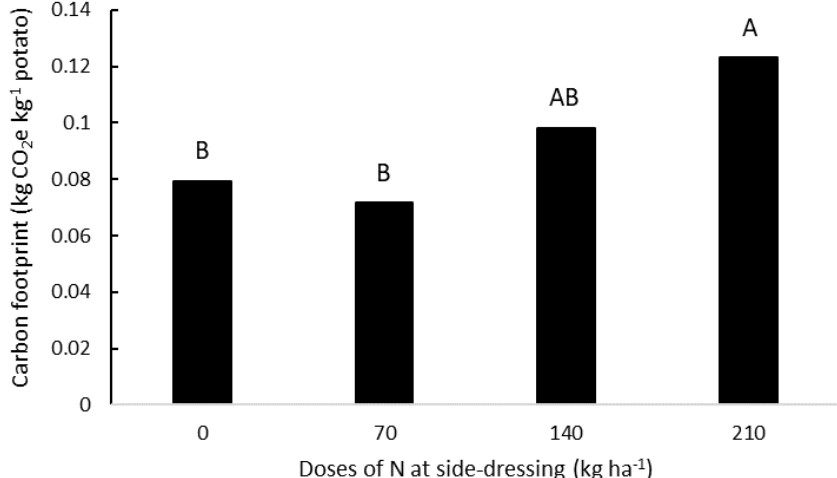

**Figure 8.** Effects of nitrogen fertilization at side-dressing on carbon footprint. Different letters above the bars indicate statistical differences between treatments, according to a Tukey test ($p \leq 0.05$).

## 4. Discussion

The results observed in this study indicate that the impact of N fertilization on potato growth and quality is contingent upon the N source and dosage applied. Regardless of the chemical form and amount of N administered, potato plants displayed N and K contents below the recommended range for optimal crop growth, as proposed by Lorenzi et al. [19]. However, despite this deviation, no visual symptoms were observed, and the plants achieved yields surpassing the Brazilian national average of 31.7 t ha$^{-1}$ [28].

N application rates and sources promoted diverse effects on gas exchange parameters [29,30]. The application of rates exceeding 110 and 91 kg N ha$^{-1}$ for stomatal conductance and net photosynthetic rate, respectively, coupled with lower yields, indicates that an increase in N supply beyond the optimal rate disrupts physiological mechanisms, manifesting in diminished tuber biomass production. Stomatal conductance decreased as N application rates increased, especially when surpassing crop demand, leading to a reduction in transpiration rate and forming a barrier to the uptake of atmospheric $CO_2$. Consequently, diminished internal $CO_2$ concentration contributed to a decrease in net photosynthetic rate, ultimately affecting potato plant yields [31]. These outcomes can be elucidated from two perspectives. Firstly, they may be associated with increments in N-compounds at the expense of starch accumulation, promoting vigorous vegetative growth. This resulted in an increase in the shoot/root ratio, hindering the development of the root system and reducing the plant's resistance to dry periods through stomatal closure [32]. Secondly, lower K contents in potato plants could also contribute, influencing the optimal turgor of stomatal guard cells [33].

The application of rates greater than 116 kg N ha$^{-1}$ reduced N use efficiency, affecting both potato development and dry mass accumulation. The results observed in this study are consistent with those reported by several other authors, who also noted a decrease in N use efficiency as N supply increased [34–36].

Notably, differences in potato production were noticed between fertilization with ammonium nitrate and urea. Potato plants accumulated more nitrate when fertilized with ammonium nitrate than with urea. This difference arose because ammonium nitrate supplies N in the forms of nitrate ($NO_3^-$) and ammonium ($NH_4^+$), whereas urea primarily provides N in the form of urea ($CO(NH_2)_2$), which is not readily available to plants. Urea must first be converted by soil microbes into $NH_4^+$, and subsequently into $NO_3^-$, before plants can take it up. In this case, $NH_4^+$ is readily taken up by plant roots and directly assimilated into plant proteins and other organic compounds. $NO_3^-$, on the other hand, must be converted to $NH_4^+$ before it can be used by potato plants. Given that ammonium nitrate already contains N in the form of $NO_3^-$, it offers a more readily available source of nitrate for plants. Consequently, when fertilized with ammonium nitrate, plants can absorb nitrate more quickly, resulting in greater nitrate accumulation in plant tissues compared to urea fertilization [29].

Urea was the most suitable source for reducing nitrate content in tubers while maximizing potato yield. The optimal treatment, resulting in the highest yield, involved the application of 136 kg N ha$^{-1}$ urea as the N source, which resulted in a nitrate content equivalent to 49.24 mg kg$^{-1}$, below the recommended limit for human consumption (200 mg $NO_3^-$ kg$^{-1}$ of fresh mass). These findings can be attributed to the immediate availability of nitrate in the soil to plants when ammonium nitrate (composed of 34% of N—50% $NO_3^-$ and 50% $NH_4^+$) is supplied. Elevated levels of ammonium can be detrimental to plants, negatively impacting the uptake of essential cations such as $K^+$, $Mg^{2+}$, and $Ca^{2+}$. Additionally, these high levels can lead to changes in cell pH and result in ionic and hormonal imbalances, thereby affecting photosynthetic rates, plant development, biomass production, and yield [37,38].

Although there are varying standards worldwide regarding the maximum permissible levels of nitrate in vegetable crops, Brazil has not yet established its own standard. However, regardless of the sources and rates applied, the nitrate contents recorded in this study were within the limits established by German standards, which permit levels up to

200 mg $NO_3^-$ $kg^{-1}$ of fresh mass [39]. Regulating nitrate concentrations is crucial. It is essential to recognize that the acceptable limit of nitrate in each food item varies based on its consumption quantity and type. While potatoes are considered low-nitrate crops, their extensive consumption in many regions worldwide can lead to substantial nitrate intake [40]. This raises health concerns, as nearly 80% of the total $NO_3^-$ intake of the human population comes from the ingestion of potatoes and other vegetables [12].

The boiling method significantly reduced nitrate content in potato tubers. Since nitrate is highly soluble in water, the boiling process enhances its diffusion into the cooking water, thereby reducing its content in the tuber [41]. This finding suggests that cooking tubers through boiling can be an effective method to deplete nitrate levels, thereby reducing associated health risks. Considering a daily consumption of 88.5 g of potato [42] and a nitrate content of 49.24 mg $kg^{-1}$ FM (nitrate content obtained at 136 kg N $ha^{-1}$ from urea—highest yield), the daily intake of nitrate would be 4.4 mg $NO_3^-$ $kg^{-1}$ in raw tubers and 2.4 in boiled tubers.

In light of the absence of established standards and legislation regarding nitrate levels in potato crops in Brazil, it is imperative to address this regulatory gap to safeguard public health and ensure food safety. Implementing specific limits or regulations on nitrate concentrations in potatoes would provide clarity and guidance for farmers, retailers, and consumers alike. These standards could be tailored to reflect consumption patterns and dietary habits unique to Brazil, taking into account regional dietary preferences and preparation methods that could help reduce tuber nitrate content. Therefore, the findings of this study hold significant potential to inform the development of nitrate legislation in Brazil. By providing valuable insights into nitrate levels in potato crops and their implications for public health, this research can serve as a foundation for the establishment of targeted regulations and standards. Determining the proper N source and dose for potato plants not only benefits the yield and quality of tubers, but also reduces the total operational costs of the system [43]. Based on an average price of US$699.38 per ton of urea [44], the cost of applying 136 kg N $ha^{-1}$ (US$95.12) would be 193.4% lower than applying the same dose of ammonium nitrate (US$278.93).

Greenhouse gas emissions, specifically $N_2O$, tend to increase proportionally with the doses of N applied. However, when analyzing the carbon footprint, the lowest value was obtained at an application rate of 70 kg N $ha^{-1}$, approximately 41% lower than the highest value observed with 210 kg N $ha^{-1}$ and 26% lower than with 140 kg N $ha^{-1}$ (Figure 8). These differences are directly linked to potato yield: when 70 kg N $ha^{-1}$ is used, it is possible to obtain yields similar to those obtained when the highest doses of N are applied. This increased efficiency in N utilization by potato plants not only contributes to climate benefits by reducing the carbon footprint, but also promotes sustainable agriculture practices [37].

## 5. Conclusions

The findings of this study reveal that screening sources and rates of N are crucial for developing an effective potato nutrition program. Proper management of N fertilization emerges as a key determinant of agronomic outcomes and tuber quality, significantly influencing both. Consistent with the hypotheses, the optimal strategy for maximizing profits in the agricultural sector without compromising tuber quality involves the application of 136 kg N $ha^{-1}$ using urea as the N source, which results in a nitrate content equal to 49.24 mg $kg^{-1}$, below the recommended limit for human consumption (200 mg $NO_3^-$ $kg^{-1}$). Additionally, the analysis reveals that the lowest carbon footprint was obtained when 70 kg N $ha^{-1}$ was applied, around 41% and 26% lower than when 210 kg N $ha^{-1}$ and 140 kg N $ha^{-1}$ was applied, respectively. These results provide compelling evidence supporting the hypothesis that precise calibration of N doses alongside appropriate sourcing can effectively mitigate over-fertilization in potato cultivation while concurrently enhancing agronomic performance and minimizing environmental impact.

**Author Contributions:** Conceptualization, A.B.C.F.; methodology, A.B.C.F.; software, C.S.N. (Camila Seno Nascimento) and B.d.J.P.; validation, A.B.C.F., C.S.N. (Camila Seno Nascimento) and B.d.J.P.; formal analysis, C.S.N. (Camila Seno Nascimento), C.S.N. (Carolina Seno Nascimento) and B.d.J.P.; investigation, C.S.N. (Camila Seno Nascimento), C.S.N. (Carolina Seno Nascimento) and B.d.J.P.; resources, A.B.C.F.; data curation, C.S.N. (Camila Seno Nascimento), C.S.N. (Carolina Seno Nascimento), B.d.J.P., P.H.S.S., M.C.P.d.C. and A.B.C.F.; writing—original draft preparation, C.S.N. (Camila Seno Nascimento), C.S.N. (Carolina Seno Nascimento). and B.d.J.P.; writing—review and editing, A.B.C.F.; visualization, A.B.C.F.; supervision, A.B.C.F.; project administration, A.B.C.F.; funding acquisition, A.B.C.F. and C.S.N.(Camila Seno Nascimento). All authors have read and agreed to the published version of the manuscript.

**Funding:** This research received no external funding.

**Data Availability Statement:** Data are contained within the article.

**Acknowledgments:** The authors are grateful to CNPq and CAPES.

**Conflicts of Interest:** The authors declare no conflicts of interest.

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
