# Peer review of "Enhancing Sustainability in Potato Crop Production: Mitigating Greenhouse Gas Emissions and Nitrate Accumulation in Potato Tubers through Optimized Nitrogen Fertilization"

_nitrogen, doi:10.3390/nitrogen5010011_

Round 1
Reviewer 1 Report
Comments and Suggestions for Authors
Dear Authors,As a reviewer, I have the following comments on the submitted text:
ad. 1. The abstract lacks research hypotheses. They should be put there.
ad. 2. The conclusions should refer more clearly to the hypothesis - confirmed or refuted and why.
ad. 3. Both the literature list and the text should only use the latest publications, i.e. not older than 2010, because this type of research is still being continued in the literature on the subject. For example, the following literature items should be replaced with new ones: 9; 12; 13; 14; 15; 16; 17.
ad. 4. The list of literature used in the text should be increased by at least 10-15 items, e.g. in the discussion or introduction.
Other parts of the article are written correctly. Methods well used. Results well described.
After taking into account my comments, the article can be published.
Regards
Reviewer 2 Report
Comments and Suggestions for Authors
Dear Authors,
Below my suggestions:
Introduction: You should write something about the importance of potato in current agriculture, its normal impact on GHG emissions and detail better the goals of the research, adding more explanations and information about the type of N engaged in this research and doses.
Lines 66-68: Specify the source where you too this information (weather station or literature).
Lines 69-70: Why did you test pH in calcium chloride?
Lines 69-73: The work needs carbon or soil organic matter values. It has no sense investigate deeply N dynamics if you do not know the soil organic condition.
Lines 113-114: Provide more info about WUE calculation with reference.
Lines 115-116: Extraction in acetone? Specify.
Lines 126-127: Provide more info about the method.
Lines 129-134: I think you should introduce this method previously.
Lines 143-147: Provide more info about the calculation (e.g. formulas, equations..)
Lines 166-169: Report all the statistical analyses that you tested. Any Tukey’s test, ANOVA or other? What about the total number of replicates for sampling and lab measurements?
Lines 270-273: What about chl a e b alone? And what about chl a/b? Report these parameters as they are very informative for the photosynthetic response of the plant to nitrogen treatment, acting as rapid responses in photosynthetic change and treatment.
Discussion: Do not use acronyms for parameters and do not repeat results (numbers) in this section.
Lines 548-549: Is Germany the only one to have limits? What about USA or other countries of Latin America?
The discussion should provide possible solutions in terms of limits or legislation that Brazil does not have so far.
Round 2
Reviewer 2 Report
Comments and Suggestions for Authors
Dear authors, the work is now improved. Thank you.